# Pre-transplant residual diuresis and oxalic acid concentration influence kidney graft survival

Gideon Post Hospers[1]*, Mirjam Laging[1], Wesley J. Visser[2], Pedro Miranda Afonso [ID][3,4], Jeroen GHP Verhoeven[1], Ingrid RAM Mertens zur Borg[5], Dennis A. Hesselink[1], AnnekeM.E. de Mik-van Egmond[2], Michiel G.H. Betjes[1], Madelon van Agteren[1], David Severs[1], Jacqueline van de Wetering [ID][1], Robert Zietse[1], Michel J. Vos[6], Ido P. Kema[6], Marcia M.L. Kho[1], Marlies E.J. Reinders[1], Joke I. Roodnat [ID][1]

1 Department of Internal Medicine, Department of Nephrology and Transplantation, Transplant Institute, Erasmus Medical Center, Rotterdam, The Netherlands, 2 Department of Internal Medicine, Division of Dietetics, Erasmus Medical Center, Rotterdam, The Netherlands, 3 Department of Biostatistics, Erasmus Medical Center, Rotterdam, The Netherlands, 4 Department of Epidemiology, Erasmus Medical Center, Rotterdam, The Netherlands, 5 Department of Anaesthesiology, Erasmus Medical Center, Rotterdam, The Netherlands, 6 Department of Clinical Chemistry Metabolic diseases, University Medical Center Groningen, The Netherlands

* gideon-posthospers@hetnet.nl

## Abstract

### Background and hypothesis

Oxalic acid, a toxic metabolic end product, accumulates when kidney function deteriorates. Apart from its direct tubulotoxicity, it crystallizes at concentrations above 30–40 µmol/L. High oxalic acid concentrations at transplantation might negatively influence kidney transplant function. The influence of the concentrations of oxalic acid and its precursors and residual diuresis on kidney transplant outcomes was studied.

### Methods

In this prospective cohort study, patients who received a kidney transplant between September 2018 and January 2022 participated. Concentrations of oxalic acid and precursors were determined in pre-transplant blood samples. Data on residual diuresis and other recipient, donor or transplant related variables were collected. Follow-up lasted until July 1st 2023.

### Results

496 patients were included, 154 were not on dialysis. Median residual diuresis was 1000 mL/day (IQR 200; 2000 mL/day). There were 230 living donor transplantations. Oxalic acid concentrations exceeded the upper normal concentration in 99% of patients, glyoxylic acid in all patients. There were 52 (10%) graft failures. As the influence of oxalic acid on the risk of graft failure censored for death was non-linear, it was categorized into two groups: ≤ 60 and > 60 µmol/L. In multivariable Cox analysis

**Data availability statement:** Our data is pseudonymized data, which, in the Netherlands, falls under the General Data Protection Regulation (GDPR). The raw data can be made available on request, under the conditions of a Data Transfer Agreement (DTA). Data requests can be sent to the first (gideon-posthospers@hetnet.nl), the second (m.laging@erasmusmc.nl) and last author (j.roodnat@erasmusmc.nl), but also to a non-author institutional point of contact: data-managers.niertransplantatie@erasmusmc.nl.

**Funding:** Funding: Foundation "Stichting de Merel" The funders had no role in study design, data collection and analysis, decision to publish, or preparation of the manuscript.

**Competing interests:** The authors have declared that no competing interests exist.

the graft failure censored for death risk was significantly influenced by residual diuresis, donor type (living versus deceased), donor age and oxalic acid. In 180 patients oxalic acid concentration shortly after transplantation was significantly lower than pre-transplant concentrations, suggesting excretion by the new graft. A better eGFR at day 7 was associated with lower oxalic acid concentration. Oxalic acid and residual diuresis did not influence patient survival.

## Conclusion

Residual diuresis and oxalic acid concentration are important and independent predictors of graft survival censored for death. These results underline the importance of pre-emptive transplantation, or optimizing the pre-transplant patients' condition regarding waste product concentrations.

## Introduction

Kidneys are superior to dialysis in removing various (non-urea) solutes and even clinically negligible residual kidney function has been shown to provide non-urea solute clearance [1–5]. Toxic waste products accumulate in patients with (end stage) kidney disease, especially when diuresis is reduced [6,7].

Oxalic acid is one of these toxic waste products that are primarily excreted via glomerular filtration and tubular secretion by the kidney [8,9]. It is the end product of many metabolic processes in the body. Elgstoen et al. showed a positive relationship between serum creatinine and plasma oxalic acid concentrations in patients without primary or enteric hyperoxaluria [10]. Although dialysis can remove oxalic acid, concentrations typically rebound to pre-dialysis concentrations within 48 hours [11,12].

The super saturation threshold concentration of oxalic acid is supposed to be 30–40 µmol/L [13]. When concentrations in urine are above the threshold, calcium oxalate precipitates in the kidney [14,15]. When plasma concentrations are above the threshold, calcium oxalate precipitates in other organs as well [16–21]. There is a high oxalate nephropathy (recurrence) rate after kidney transplantation in patients with primary or enteric hyperoxaluria [14,22,23]. Kidney patients without primary or enteric hyperoxaluria may also have high oxalic acid concentrations, caused by the failing excretion. Previous studies in this population have shown that calcium-oxalate deposition in kidney biopsies within three months after transplantation heralds a poor prognosis regarding graft function [24,25]. In addition to precipitation, oxalic acid also demonstrates tubulotoxicity and may cause inflammation contributing to progressive kidney injury, even in the absence of visible precipitates [26–29].

Two studies found an increased mortality risk in kidney patients with high oxalic acid concentrations: Diabetic dialysis patients with high plasma oxalic acid concentrations were found to have a significantly increased mortality risk [30]. In another study, transplant recipients with high oxalic acid concentration at ten weeks after transplantation exhibited increased mortality rates, while the graft failure risk barely missed significance [31].

Apart from oxalic acid, its precursors, such as glycoaldehyde, glycolic acid, glyoxylic acid and glyceric acid, may accumulate as well when kidney function deteriorates (Fig 1) [32]. Kidney toxicity of these precursors has been studied as they also are metabolites of ethylene glycol (Fig 1). Ethylene glycol poisoning is notorious for its toxicity after ingestion [29,33]. Ethylene glycol itself is not toxic, but it enters the endogenous oxalate synthesis pathway via breakdown to glycoaldehyde and eventually to oxalic acid. The metabolites oxalic acid, glyoxylic acid and glycoaldehyde were shown to be tubulotoxic [26,28] and may lead to acute kidney failure. Very high concentrations may also lead to neurotoxicity and death if untreated [34]. In the kidney transplant setting the graft is placed in an environment with high concentrations of all kinds of waste products.

Low residual kidney function is associated with an increased delayed graft function (DGF) risk after kidney transplantation [35–37]. It cannot be ruled out that the effect that is attributed to low residual diuresis, in reality is the effect of a whole collection of accumulated toxic waste products. Therefore, the influence of residual diuresis, and oxalic acid and its precursors was studied in the same model. This study aimed to examine the influence of waste products as oxalic acid and its precursors and residual diuresis on kidney transplant outcomes.

## Materials and methods

All adult patients referred for kidney transplant work-up between September 1st 2018 and January 1st 2022 were invited to participate in this study. Follow-up lasted until July 1st 2023. The study conformed to the principles outlined in the Declaration of Helsinki, and was approved by the Medical Ethics Committee of Erasmus University Medical Center (EMC) Rotterdam [MEC 2018–044]. All patients provided written informed consent before inclusion. Participation comprised transplantation in the period studied, and a ten mL blood sample taken in the operating room immediately before transplantation. Concentrations of oxalic acid and its precursors: glyoxylic acid, glycolic acid and glyceric acid were determined. The volume of residual diuresis per day was self-reported, and based on last 24 hour urine sample handed in for general renal care. Collection was during the 24 hour preceding a haemodialysis session or a visit to the outpatient department for peritoneal dialysis or, for pre-emptive patients, the nephrology outpatient clinic. Besides, patients were asked to complete a questionnaire about dietary habits. Results of this food frequency questionnaire are described separately.

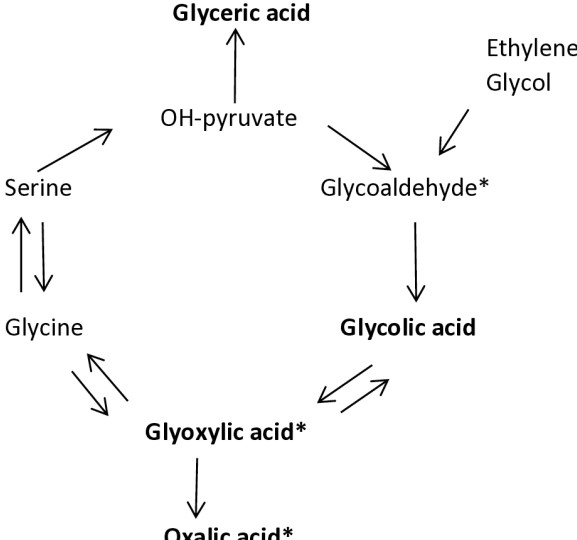

**Fig 1. Endogenous oxalic acid synthesis pathway.** In **bold** are the substances whose concentrations were determined. * Known to be nephrotoxic, tubulotoxic.

All pre transplant survival and kidney function related recipient, donor and transplant characteristics supposed to influence transplant results were collected in the database and included in the analysis.

There was no post transplantation protocol time for blood collection for determination of oxalic acid and precursor concentrations. In patients scheduled for a graft biopsy within 3 months after transplantation, blood was taken on the day of the biopsy for determination of serum concentrations of oxalic acid and its precursors. Almost all these blood collections were performed within 2 weeks after transplantation (median 7 (IQR 5–10 days).

### Laboratory tests

To quantify the plasma organic acids (oxalic acid, glyoxylic acid, glycolic acid and glyceric acid), blood was taken and placed on ice followed by centrifugation at 4°C without delay. Heparinised plasma samples were de-proteinized by addition of 75 µL 37% hydrochloric acid to 0.5 mL plasma followed by centrifugation. The supernatant was stored at -70°C until analysis. For quantification, a gas chromatography mass spectrometry method (GC-MS) was used that was developed and validated according to the guidelines for bioanalytical method validation of the Dutch Scientific Association of Medical Laboratory Specialists in Clinical Chemistry and the ISO 15189:2012 standard describing requirements for quality and competence in medical laboratories. The current method is a further development of previous methods, as an in-house developed method and fulfils all requirements stated in Regulation (EU) 2017/746 of the European Parliament and of the Council on in vitro diagnostic medical devices for human use [38,39]. In short, the organic acids were derivatized using ethyl chloroformate combined with 1-propanol and pyridine followed by extraction with heptane. For separation a 15 meter ZB-FFAP column (Phenomenex) was used followed by quantification (Agilent 7890B GC) in positive chemical ionization mode using a gas reaction mixture of 5% ammonia in methane. Stable isotope labelled internal standards used were either 13C labelled (oxalic acid, glycolic acid, glyoxylic acid) or deuterium labelled (glyceric acid).

The precision of the assay was evaluated by measuring internal quality control (QC) samples at three concentration levels: low (L), medium (M), and high (H). The coefficients of variation (CV%) were determined for oxalate (oxalic acid), glycolate (glycolic acid), glycerate (glyceric acid), and glyoxylate (glyoxylic acid). The results are summarized as follows: oxalate CV% of 3.16 (L), 3.95 (M), and 4.24 (H); glycolate CV% of 2.52 (L), 1.87 (M), and 1.36 (H); glycerate CV% of 7.36 (L), 9.77 (M), and 5.98 (H); glyoxylate CV% of 6 (L), 4.14 (M), and 2.77 (H).

The sensitivity of the assay is reflected in the Lower Limit of Quantitation (LLOQ), which was determined to be 0.50 µmol/L for oxalic acid. The assay demonstrated excellent precision, with intra-assay, inter-assay, and repeatability coefficients of variation (CVs) all below 10%, meeting predefined acceptance criteria. The recovery of the method ranged between 88–113%, depending on matrix conditions, ensuring reliable quantification.

Regarding specificity, the GC-MS method provides high analytical specificity due to the use of selective ion monitoring, with oxalic acid quantified at m/z 191.9. The method validation confirmed no significant carry-over (<0.1%) and established a linear range up to 159 µmol/L, with a correlation coefficient ($r^2$) of 0.9997.

### Data analysis

Statistical analysis was performed with IBM SPSS Statistics version 24 and R version 4.2.1 using the survival packages (v. 3.5.5). Baseline characteristics and outcomes were described as counts and percentages for categorical variables assessed using the Chi-square test. Continuous skewed variables were expressed as medians and interquartile ranges (IQR) and analysed via the Mann-Whitney U-test. A two sided p-value less than 0.05 was considered statistically significant.

Uni- and multivariable Cox proportional hazards analyses were performed to identify risk factors for death-censored graft survival and patient mortality using preselected variables (Table 1). Graft failure is defined as need to resume dialysis and/or a non-functioning graft on a nuclear scan. Natural cubic splines and polynomial functions were used to evaluate nonlinear effects of oxalic acid, residual diuresis, and donor age [40]. To determine the maximum number of degrees of

**Table 1. Recipient, donor and transplantation characteristics from the entire population, the population of recipients with a living donor and the population with a deceased donor kidney. PD: peritoneal dialysis; DBD: donation after brain death; DCD: donation after cardiac death; HD: hemodialysis; HLA: human leukocyte antigen; IQR: interquartile range; vPRA: virtual Panel Reactive Antibodies.**

| | Total population N=496 | Living donor n=230 | Deceased donor n=266 | p-value living versus deceased donor |
|---|---|---|---|---|
| Recipient characteristics: | | | | |
| Gender (male) n *(%)* | 300 (60.5) | 131 (57) | 169 (64) | 0.08 |
| Age (years) *median (IQR)* | 62 (51; 69) | 58 (46; 66) | 65 (55; 71) | <0.001 |
| BMI (kg/m2) *median (IQR)* | 27 (24; 31) | 26 (23; 29) | 28 (24;33) | <0.001 |
| CRP (mg/L) *median (IQR)* | 3 (1; 8) | 2 (1;5) | 5 (2;11) | <0.001 |
| Medical history: | | | | |
| Cardiac event n *(%)* | 89 (17.9) | 30 (13) | 59 (22) | 0.005 |
| Cerebrovascular accident n *(%)* | 60 (12.1) | 24 (10) | 36 (14) | 0.18 |
| Vascular event n *(%)* | 43 (8.7) | 12 (5) | 31 (12) | 0.008 |
| Diabetes mellitus n *(%)* | 167 (33.7) | 49 (21) | 118 (44) | <0.001 |
| Residual diuresis (mL/day) *median (IQR)* | 1000 (200; 2000) | 1550 (737; 2000) | 500 (29; 1500) | <0.001 |
| Use of diuretics, yes n *(%)* | 184 (37.1) | 83 (36) | 100 (38) | 0.499 |
| Dialysis n *(%)* | | | | <0.001 |
| No | 154 (31.0) | 126 (55) | 28 (11) | |
| PD | 105 (21.2) | 43 (19) | 62 (23) | |
| HD | 237 (47.8) | 61 (27) | 176 (66) | |
| Time on dialysis (months) *median (IQR)* | 15(0; 30) | 0 (0; 15) | 23 (14; 40) | <0.001 |
| Time between last dialysis and transplantation (days) | | | | |
| PD only *median (IQR)* | 0.39 (0.21; 0.71) | 0.42 (0.25;1.1) | 0.34 (0.18; 0.62) | 0.027 |
| HD only *median (IQR)* | 1.34 (0.80; 1.95) | 1.2 (1.0; 1.9) | 1.14 (0.66; 1.98) | 0.909 |
| Hyperoxaluria, non-renal cause n *(%)* | 20 (4.0) | 10 (4) | 10 (4) | 0.457 |
| Oxalic acid (µmol/L) *median (IQR)* | 33.2 (18.0; 56.6) | 22 (13; 42) | 42 (26;62) | <0.001 |
| Oxalic acid >60 µmol/L n *(%)* | 113 (22.8) | 36 (16) | 77 (29) | <0.001 |
| Glycolic acid (µmol/L) *median (IQR)* | 5.7 (5.0; 6.7) | 5.3 (4.7; 6.1) | 6 (5.2; 7.0) | <0.001 |
| Glyoxylic acid (µmol/L) *median (IQR)* | 2.0 (1.4; 2.8) | 1.7 (1.1; 2.5) | 2.1 (1.5; 3.0) | <0.001 |
| Glyceric acid (µmol/L) *median (IQR)* | 2.6 (2.2; 3.1) | 2.4 (2.0; 2.7) | 2.8 (2.4; 3.3) | <0.001 |
| vPRA *median (IQR)* | 4 (0; 5) | 4 (0; 5) | 4 (0; 13.5) | 0.692 |
| vPRA n *(%)* | | | | 0.967 |
| <4 | 229 (46.2) | 106 (46) | 123 (46) | |
| 4-84 | 231 (46.4) | 108 (47) | 123 (46) | |
| ≥85 | 36 (7.5) | 16 (7) | 20 (8) | |
| First kidney transplantation n *(%)* | 421 (84.9) | 203 (88) | 218 (82) | 0.033 |
| Donor characteristics | | | | |
| Donortype | | | | <0.001 |
| Living donor *n (%)* | 230 (46.4) | 230 (100) | | |
| DBD n *(%)* | 88 (17.7) | | 88 (33) | |
| DCD n *(%)* | 178 (35.9) | | 178 (67) | |
| Age donor (years) *median (IQR)* | 58 (48; 67) | 56 (47; 64) | 60 (50; 68) | <0.001 |
| Donor gender (male) n *(%)* | 253 (51.0) | 114 (50) | 139 (52) | 0.306 |
| Donor BMI (kg/m²) median (IQR) | 26 (23; 29) | 26 (24; 29) | 26 (23; 28) | 0.262 |

*(Continued)*

**Table 1.** (Continued)

|  | Total population N=496 | Living donor n=230 | Deceased donor n=266 | p-value living versus deceased donor |
|---|---|---|---|---|
| Donor comorbidity: |  |  |  |  |
| hypertension n (%) | 155 (31.3) | 56 (24) | 99 (37) | <0.001 |
| Diabetes mellitus n (%) | 23 (4.6) | 0 (0) | 23 (9) | <0.001 |
| Donor creatinine (μmol/L) median (IQR) | 72 (59; 83) | 77 (69; 84) | 65 (53; 80) | <0.001 |
| Transplantation characteristics |  |  |  |  |
| HLA A, B, DR mismatches |  |  |  | 0.145 |
| 0-3 | 296 (59) | 131 (57) | 165 (62) |  |
| 4-6 | 200 (40.3) | 99 (43) | 101 (38) |  |
| HLA mismatches median (IQR) | 3 (3; 6) | 3 (2;5) | 3 (2; 4) | 0.073 |
| Cold ischemia time (min) | 389 (120; 707) | 118 (105;136) | 695 (524; 808) | <0.001 |

freedom in the multivariable model, we used 10% of the number of events. Covariates were removed using backward elimination. With log minus log plots, violations of the proportional hazards assumption were evaluated. Patients were censored at the date of the last clinical follow-up. Survival probability curves were then obtained using the Breslow estimator. To detect multicollinearity between residual diuresis and oxalic acid the variance inflation factor (VIF) was tested in linear regression analysis.

## Results

There were 512 patients that consented and underwent a kidney transplantation. In 16 patients concentrations of oxalic acid and/or its precursors were missing. Results of 496 patients were available for analysis. Table 1 shows baseline recipient, donor and transplant related characteristics, there were no missing values in these 496 patients. Characteristics of the recipient population of a living donor versus recipients of a deceased donor kidney are shown in Table 1. Supplementary table 1 shows characteristics of the population of pre-emptive patients versus the population on dialysis. There are significant differences; mostly unfavorable for the latter (supplementary Table 1). There were 19 patients with enteric hyperoxaluria, one patient with primary hyperoxaluria was included. The latter patient was diagnosed with kidney insufficiency and type 1 primary hyperoxaluria at the age of 63 years. He had an excellent response to pyridoxine treatment and at the age of 65, he received a kidney only transplantation with excellent results.

Fig 2 illustrates the concentrations of oxalic acid, glyoxylic acid, glycolic acid and glyceric acid in the total patient population, in the sub-population not on dialysis and in the sub-population on dialysis. Only 1.2% of the patients had pre-transplant oxalic acid concentrations within the normal range. Glycolic acid and glyceric acid concentrations were within normal values in 87% and 22% of cases, respectively. In all patients glyoxylic acid concentrations were above the normal range. Concentrations of oxalic acid and all precursors were significantly higher in the dialysis population compared to the patients not on dialysis.

### Graft survival censored for death

Until July 2023, there were 52 (10.5%) patients with graft failure and 48 (9.7%) patients died. The cubic spline Cox regression model showed a linear relationship between residual diuresis and the log-hazard ratio of graft failure censored for death (supplementary Figs 1 and 2). Residual diuresis was included as a linear continuous variable. However, the effect of oxalic acid on graft survival turned out to be non-linear, as shown in Figs 3 and 4. The influence was not significant at low values, but became significant when the lower band of the 95% confidence interval exceeds zero. This pattern was observed for both deceased and living donor kidney transplantation. Although the exact super saturation threshold of

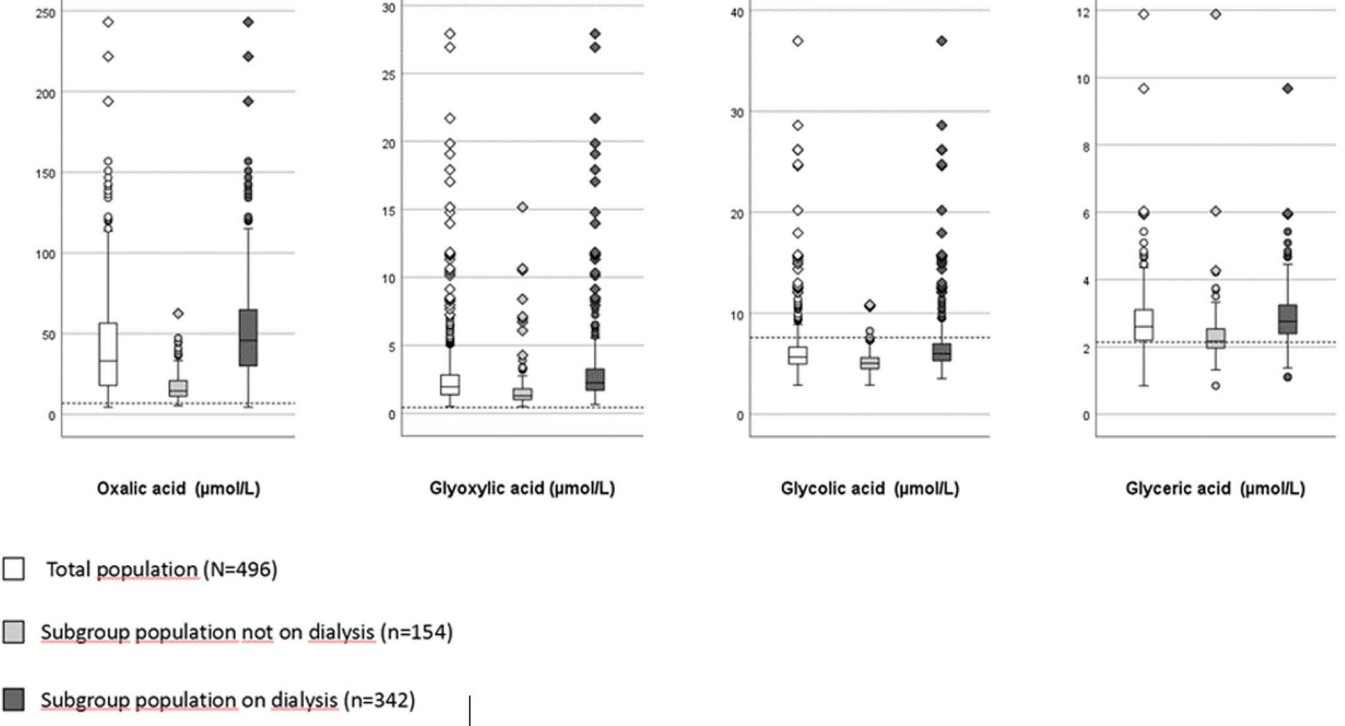

**Fig 2. Distribution of oxalic acid, glyoxylic acid, glycolic acid and glyceric acid concentrations in the population studied.** Dashed lines (-----) denote upper limits of normal concentration (oxalic acid 7.0 μmol/L; glyoxylic acid 0.4 μmol/L; glycolic acid 7.6 μmol/L; glyceric acid 2.1 μmol/L). The differences in concentrations between the subgroups on dialysis or preemptive were significant for oxalic acid and all precursors (p<0.001).

oxalic acid remains uncertain, it is generally presumed to be within the range of 30–40 μmol/L [13,14]. Above this threshold crystallisation/deposition occurs. In our population, the estimated threshold for the influence on graft survival censored for death is at 60 μmol/L in the recipient population with a living donor and at 70 μmol/L, in the recipient population with a deceased donor kidney (Figs 3 and 4). Based on Figs 3 and 4, oxalic acid concentration was categorized as ≤60 and >60 μmol/L. There were 113 patients with oxalic acid concentration>60 μmol/L; 112 of them were dialysis patients, one was pre-emptive (supplementary table 1). Consistent with previous findings, the influence of donor age followed a J-shaped curve modelled with a quadratic effect (p=0.015) [41]. Supplementary table 2 shows the results of those variables with a p-value <0.1 in univariable analysis. Though dialysis type (none, hemodialysis or peritoneal dialysis) and time between last dialysis and transplantation were significantly associated with graft failure in univariable analysis, there was no interaction between them.

All variables present in Table 1 were tested in multivariable Cox analysis. Results of the multivariable analyses with backward elimination on graft failure censored for death are shown in Table 2. Residual diuresis, donor age, donor type (living versus deceased donor kidney transplantation), and oxalic acid concentration remained in the model and significantly influenced the graft failure risk; there was no significant interaction between these variables. The proportional hazards assumption was not violated.

To detect multicollinearity between residual diuresis and oxalic acid the VIF was tested in linear regression analysis. There was no collinearity, the VIF was 1,13.

Figs 5 and 6 depict the influence of residual diuresis and oxalic acid on graft survival. For Fig 5 donor age is set to 58 years (median age) and donor type to living donor. For Fig 6 donor age is set to 58 years and donor type is set to

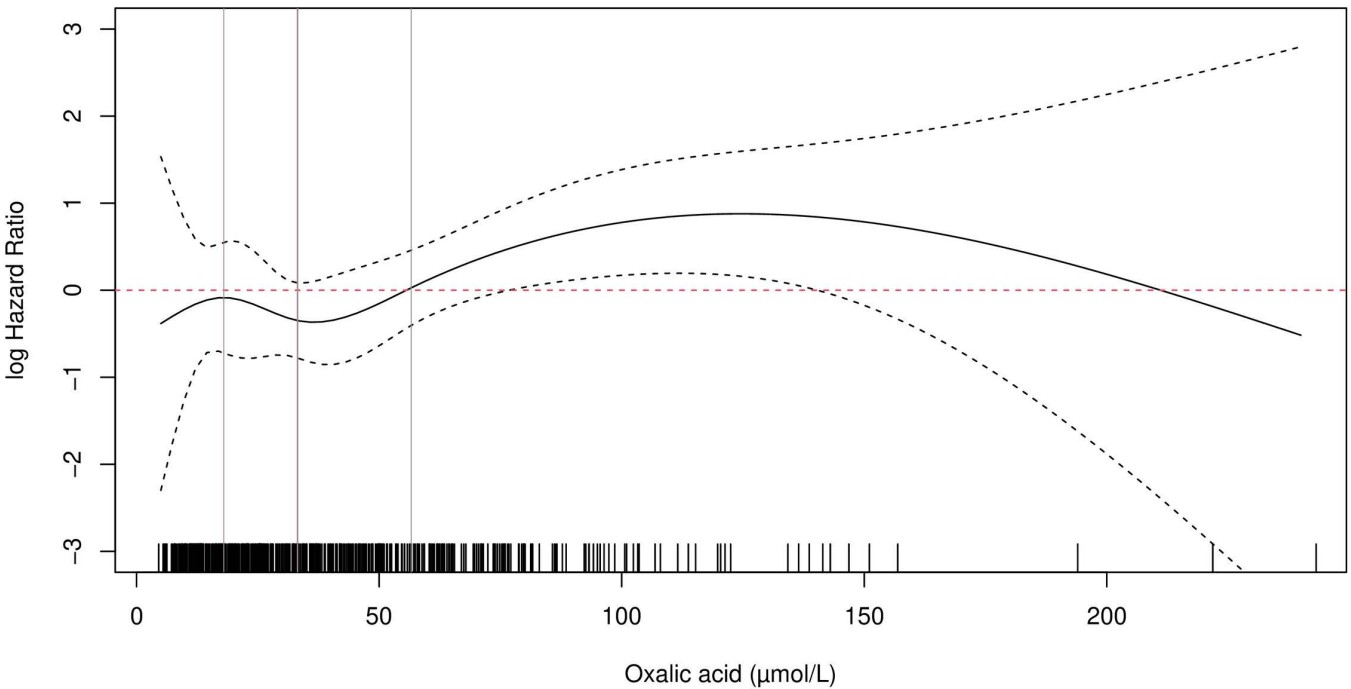

**Fig 3. The influence of oxalic acid modelled using a natural cubic spline function.** Recipients of a living donor kidney. Donor age is set to 58 years (median age), residual diuresis set to 1000mL/day (median). Median and IQR of oxalic acid concentration are shown as vertical lines. The small vertical lines above the X-axis represent the observed oxalic acid concentrations. The influence of oxalic acid on graft failure risk censored for death becomes significant when the confidence interval exceed zero.

deceased donor. Survival curves are shown for combinations of oxalic acid categories and residual diuresis volumes at the 25th and 75th percentiles.

### Patient survival

In the follow up period, 48 patients died. Oxalic acid concentration did not significantly influence the mortality risk. Variables with significant influence in univariable analysis are shown in supplementary table 3. In multivariable analysis the variables included in the final model were: recipient age, recipient CRP, glyceric acid concentration on day of transplantation, and type of kidney replacement therapy (none, hemodialysis or peritoneal dialysis) (Table 3).

### Post Transplant follow-up

From 180 patients that underwent a graft biopsy, post-transplant oxalic acid plasma concentrations were available. Almost all concentrations were obtained within 2 weeks after transplantation (median 7 (IQR 5–10) days after transplantation). In this population post-transplant plasma oxalic acid concentrations were significantly lower compared to pre transplant concentrations: mean $24 \pm 25$ versus $43 \pm 29$ µmol/L (median (IQR): 14 (7-14) versus 36 (19–59) µmol/L ($p < 0.001$)). In most patients with an eGFR $> 20$ ml/min, oxalic acid concentrations were below the upper limit of normal. Oxalic acid concentrations were still high in most patients with an eGFR $< 20$ ml/min (Fig 7). At 1 year after transplantation eGFR (estimated Glomerular Filtration Rate) was not significantly different between the patients with a functioning transplant with pre-transplant oxalic acid concentration ≤60 or $> 60$ µmol/L; respectively, $47 \pm 19$ and $43 \pm 23$ ml/min ($p = 0.079$). However, there had been more graft losses at year 1 in the population >60 (19/113; 17%) compared to the population ≤60 (16/383; 4%; $p < 0.001$).

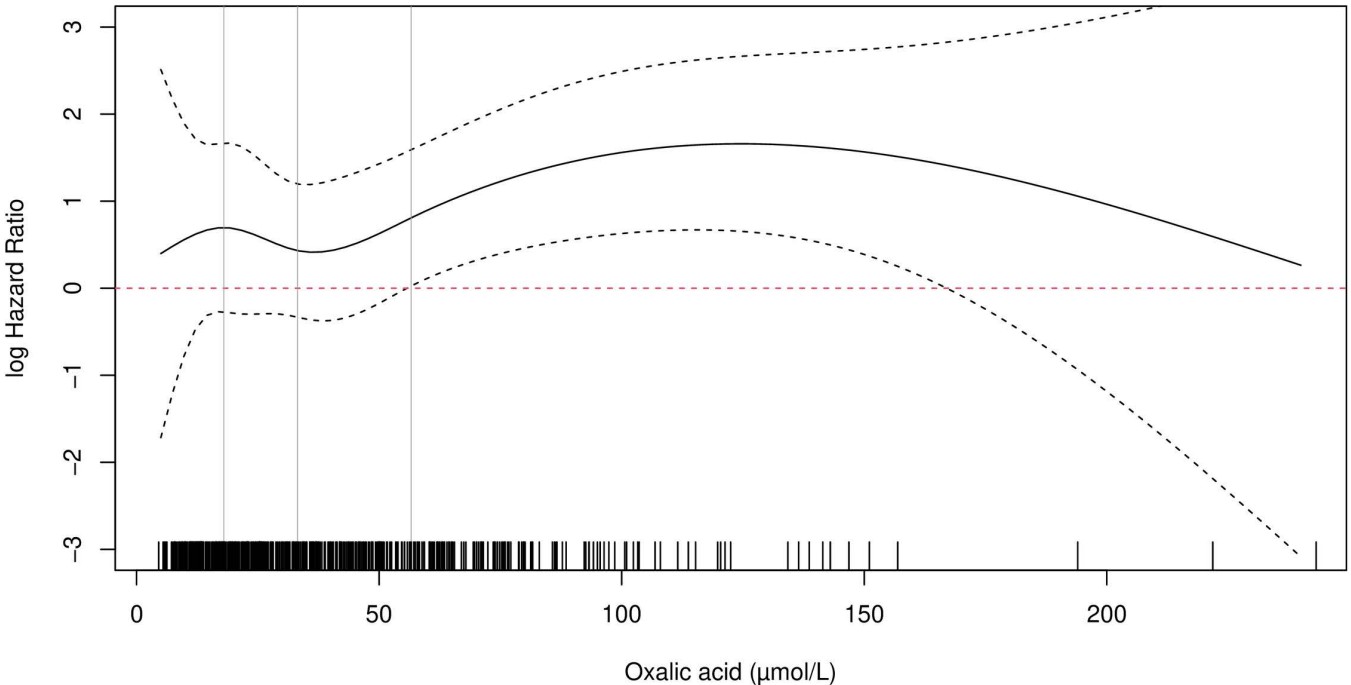

**Fig 4. The influence of oxalic acid modelled using a natural cubic spline function.** Recipients of a deceased donor kidney. Donor age is set to 58 years (median age), residual diuresis set to 1000mL/day (median). Median and IQR of oxalic acid concentration are shown as vertical lines. The small vertical lines above the X-axis represent the observed oxalic acid concentrations. The influence of oxalic acid on graft failure risk censored for death becomes significant when the confidence interval exceed zero.

**Table 2. Final model of multivariable Cox Proportional Hazards analysis on graft failure censored for death. Total population N = 496 Events N = 52. All variables listed in Table 1 were included, exclusion was performed via backward elimination.**

|  | Hazard ratio | 95% CI | | |
|---|---|---|---|---|
|  |  | Lower | Upper | Sig. |
| Residual diuresis (per 100ml) | 0.991 | 0.987 | 0.996 | <0.001 |
| Oxalic acid category (≤60 µmol/L) | 2.082 | 1.154 | 3.753 | 0.015 |
| Donor type (living) | 2.127 | 1.070 | 4.227 | 0.031 |
| Donor age (years) | 0.918 | 0.833 | 1.013 | 0.089 |
| Donor age$^2$ (years) | 1.001 | 1.000 | 1.002 | 0.024 |

For categorical variables reference values are shown in brackets. For continuous variables units are in brackets.

## Discussion

This is the first study that describes a significant effect of residual diuresis and oxalic acid concentration on the graft failure risk after kidney transplantation. We show that patients with low or absent residual diuresis, have high concentrations of more or less toxic waste products, such as glyoxylic acid and oxalic acid. It is conceivable that excretion of large amounts of these products by the newly transplanted kidney may cause toxicity and damage impairing kidney function. Even in the selected population of patients that underwent a for cause graft biopsy, post-transplant oxalic acid concentrations were significantly decreased, suggesting excretion through the graft. Figs 5 and 6 show the combined effects of residual diuresis and oxalic acid concentration on graft survival. It is remarkable that the trajectories of the lines diverge, indicating that

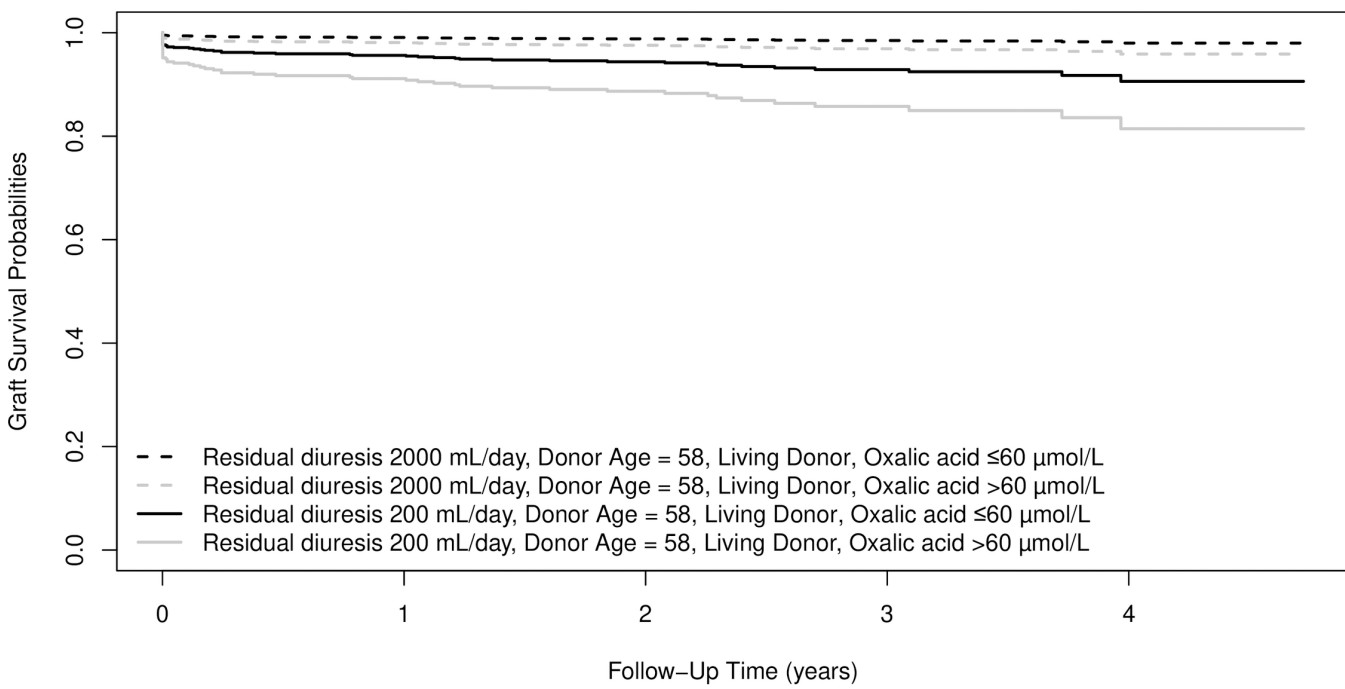

**Fig 5. Based on the results in table 2. Expected survival probabilities for four individuals who share donor age of 58 years (median donor age).** Individuals also share living donor type, but differ in residual diuresis volume (25th and 75th percentile) and in oxalic acid concentration category.

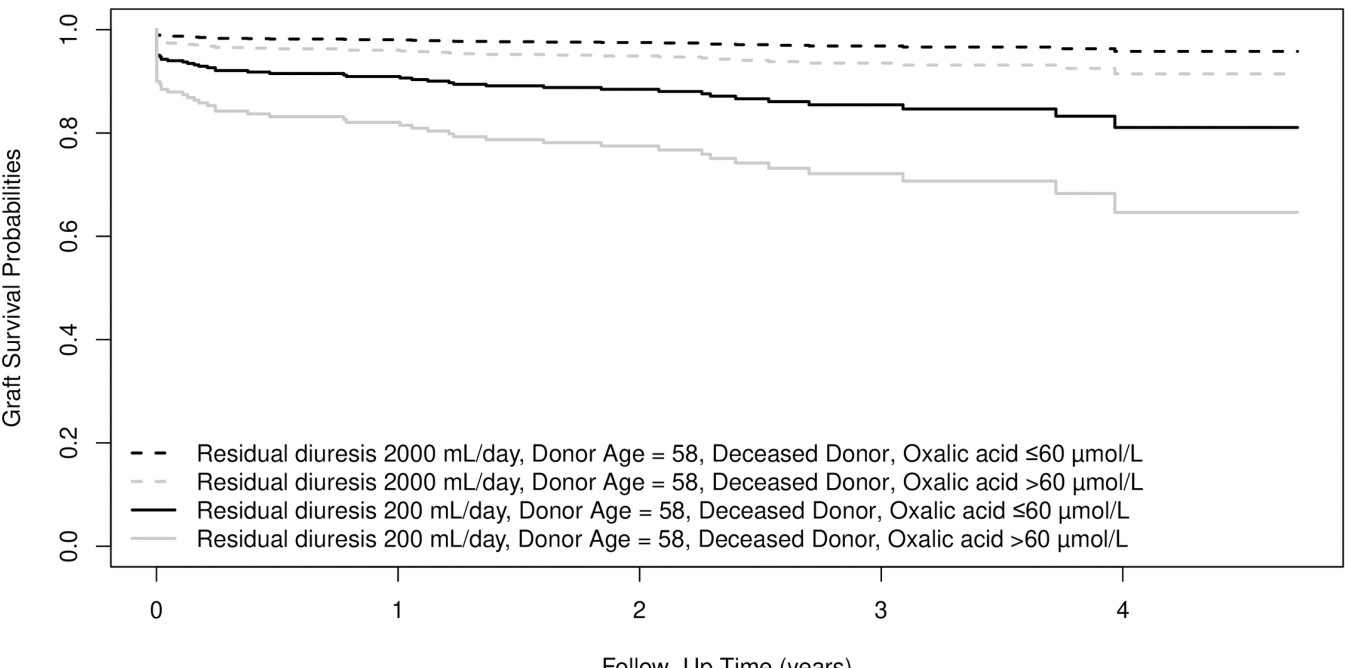

**Fig 6. Based on the results in table 2. Expected survival probabilities for four individuals who share donor age of 58 years (median donor age).** Individuals also share deceased donor type, but differ in residual diuresis volume (25th and 75th percentile) and in oxalic acid concentration category.

**Table 3. Final model of multivariable Cox Proportional Hazards analysis on patient death (N=496, events=48); All variables listed in Table 1 were included, exclusion was performed via backward elimination. The over-all effect of dialysis type as a categorical variable was tested comparing a model with and a model without dialysis type using chi-square test (P=0.024).**

| | Hazard ratio | 95.0% CI | | Sig. |
|---|---|---|---|---|
| | | Lower | Upper | |
| Recipient age (years) | 1.093 | 1.057 | 1.13 | <0.001 |
| Recipient CRP (day 0) | 1.027 | 1.007 | 1.049 | 0.01 |
| Glyceric acid concentration (µmol/L) | 1.347 | 1.061 | 1.71 | 0.014 |
| Dialysis (no) | | | | |
| Hemodialysis | 2.641 | 1.146 | 6.089 | 0.023 |
| Peritoneal dialysis | 2.994 | 1.17 | 7.677 | 0.022 |

For categorical variables reference values are shown in brackets. For continuous variables units are in brackets.

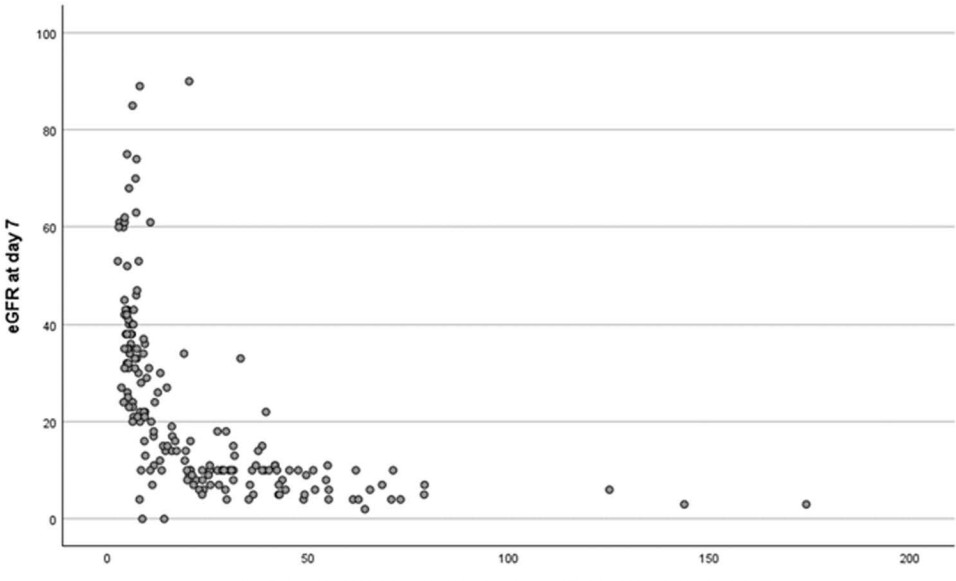

**Fig 7. Post-transplantation oxalic acid plasma concentrations, in relation to kidney function at day 7 after transplantation (n =180).**

the damage from the peri-transplantation period e.g. inflammation, toxicity and/or deposition of waste products, is only partially reversible. Figs 5 and 6 further underscore the vulnerability of recipients with low diuresis, exacerbated by high oxalic acid concentrations.

There are no studies on the influence of residual diuresis or on residual kidney function on the graft failure risk after kidney transplantation. However, we recently found that the DGF risk is significantly influenced by residual diuresis [37]. There are two other studies on the incidence of DGF, that also describe a significant influence of residual diuresis [35,36].

In patients on peritoneal dialysis [3,42] and in patients on hemodialysis [43] low residual diuresis has been shown to have a significantly negative influence on patient survival. Besides, in dialysis patients, low residual kidney function is associated with serious comorbidities e.g.: anemia, cardiovascular disease, hypertension, left ventricular hypertrophy, vascular calcifications and early atherosclerosis, inflammation and under-nutrition [27,43–49]. This means that the pre-transplant patients with low or absent residual diuresis, are less vital than those with significant residual diuresis volume. The cause

of their comorbidities is probably associated with the toxicity of waste products, left behind as a result of failing diuresis [1,4,5,50]. Sudden excretion of these products by the newly transplanted kidney may cause damage to the kidney transplant.

Our study showed that oxalic acid concentrations were above the upper normal concentration in 98.8% of pre-transplant patients. Highest concentrations of oxalic acid, glyoxylic acid, glycolic acid and glyceric acid were found in pre transplant patients on dialysis (versus not yet) and in patients with the lowest residual diuresis volume.
The influence of oxalic acid on the graft failure risk censored for death was not linear. Above a concentration of 60 µmol/L oxalic acid exerts a significant and negative effect in univariable and in multivariable analysis. Although there is no unanimity on the exact super saturation threshold of oxalic acid, it is presumed to be within the range of 30–40 µmol/L [13,14,30]. Evidently, measurable damage to the transplanted kidney occurs at concentrations higher than this threshold. The influence remained unaffected by all other variables with significant influence. This means that, although the effect of residual diuresis on graft failure probably stands for the effect of many toxic waste products, oxalic acid, on top of that, independently exerts a significant effect on the graft failure risk. Krogstad et al. studied graft and patient survival in 167 kidney transplant patients from whom an oxalic acid concentration was determined at ten weeks after transplantation [31]. Patient survival was significantly worse in patients in the highest oxalic acid quartile, while its influence on graft survival censored for death barely failed significance (p = 0.053). The oxalic acid concentrations were relatively low in their study, most probably as a result of the dramatic drop in concentrations that happens after successful transplantation.
Oxalic acid is a small molecule, that can effectively be removed by hemodialysis. Increased net secretion (in the proximal tubule) becomes apparent with high oxalic acid intake [51]. Though concentrations decrease during dialysis, normalization of concentration depends on pre dialysis values but is seldom reached. Probably these concentrations reflect oxalic acid stores in the body. In patients with primary hyperoxaluria, even current dialysis programs are not able to prevent progressive oxalate accumulation in the majority of patients [52,53]. However, also in our pre transplant population, high oxalic acid, glyoxylic acid and glyceric acid concentrations are found in the vast majority of patients, while highest concentrations are found in those on dialysis.

Finally, there was no significant association between the incidence of patient mortality and oxalic acid concentrations, but high concentrations of its precursor glyceric acid were unfavourably associated with patient survival. Two rare, autosomal recessive metabolic disorders can result in significant excretion of urinary glycerate: primary hyperoxaluria type II and D-glyceric aciduria. There are no publications on toxicity of glyceric acid in the concentrations we found in our population. A limitation of our study is that residual function (residual eGFR), in dialysis patients with residual diuresis, was not available. Residual urine volume was included instead.

In conclusion: Residual diuresis and oxalic acid concentration are important and independent predictors of kidney graft survival censored for death. Preservation of residual diuresis, even after starting dialysis, is useful and gained attention nowadays [50]. Preservation can be achieved by using incremental hemodialysis, avoiding nephrotoxic events and treatments, and blood pressure control [54]. Dietary, lifestyle and pharmacological interventions are defined to ensure optimal native kidney function preserving care [55]. However, in addition to urea clearance (Kt/V) [56], more attention should be paid to removal of other substances that may not be sufficiently cleared by dialysis treatment. Examples are oxalic acid and glyoxylic acid: both recognized as tubulotoxic agents [26,28]. Our complementary study on dietary habits in the population studied, showed a significant relationship between dietary intake of oxalic acid and plasma oxalic acid concentration in this population of patients with end stage kidney disease [57]. In our center, efforts to reduce oxalic acid concentrations in patients with enteric hyperoxaluria led to the development of the successful hyperoxaluria treatment guideline to prevent early recurrence after transplantation [58]. In this guideline oxalic acid concentrations are reduced through dietary restrictions and intensified dialysis in the period before transplantation.

Our present study adds an argument to stimulate pre-emptive transplantation in patients who still have adequate diuresis and relatively low concentrations of oxalic acid and its precursors.

## Supporting information

**S1 Table. Recipient, donor and transplantation characteristics from the entire population, the pre-emptive recipient population and the population on dialysis.**
(XLSX)

**S2 Table. Univariable Cox Proportional Hazards analysis on graft failure censored for death (N = 496).** (All variables listed in Table 1 were tested. Only those characteristics with a p-value <0.1 in univariable analysis are shown in this table).
(XLSX)

**S3 Table. Univariable Cox Proportional Hazards analysis on patient death (N = 496).** (All variables listed in Table 1 were tested. Only those characteristics with a p-value <0.1 in univariable analysis are shown in this table).
(XLSX)

**S1 Fig. The influence of residual diuresis modelled using a natural cubic spline function.** Recipients of a living donor kidney. Donor age is set to 58 years (median age), oxalic acid concentration is set to 33.2 µmol/l (median). Median and IQR of residual diuresis concentration are shown as vertical lines. The small vertical lines above the X-axis represent the observed residual diuresis volumes. The influence of residual diuresis on graft failure risk censored for death becomes significant when the confidence interval exceeds zero (dotted lines).
(TIF)

**S2 Fig. The influence of residual diuresis modelled using a natural cubic spline function.** Recipients of a deceased donor kidney. Donor age is set to 58 years (median age), oxalic acid concentration is set to 33.2 µmol/l (median). Median and IQR of residual diuresis concentration are shown as vertical lines. The small vertical lines above the X-axis represent the observed residual diuresis volumes. The influence of residual diuresis on graft failure risk censored for death becomes significant when the confidence interval exceeds zero (dotted lines).
(TIF)

## Acknowledgments

We would like to thank Mrs Judith Kal-van Gestel MSc and Mrs Tessa Royaards for their updates of regular recipient and living donor data, Mrs Ineke Tieken, MSc for providing Eurotransplant deceased donor data, Mr Chris Ramakers, MSc for sample processing and storage and Mrs Sara J Baart, PhD for her statistical advice.

## Author contributions

**Conceptualization:** Mirjam Laging, Wesley J. Visser, Anneke ME de Mik-van Egmond, Madelon van Agteren, Jacqueline van de Wetering, Marcia ML Kho, Joke I Roodnat.

**Data curation:** Gideon Post Hospers, Mirjam Laging, Wesley J. Visser, Pedro Miranda Afonso, Jeroen GHP Verhoeven, Ingrid RAM Mertens zur Borg, Anneke ME de Mik-van Egmond, Michiel GH Betjes, Madelon van Agteren, Ido P Kema, Marcia ML Kho, Joke I Roodnat.

**Formal analysis:** Gideon Post Hospers, Mirjam Laging, Pedro Miranda Afonso, Jeroen GHP Verhoeven, Ingrid RAM Mertens zur Borg, Michiel GH Betjes, Madelon van Agteren, Jacqueline van de Wetering, Robert Zietse, Ido P Kema, Marcia ML Kho, Joke I Roodnat.

**Funding acquisition:** Michiel GH Betjes, Robert Zietse, Joke I Roodnat.

**Investigation:** Gideon Post Hospers, Mirjam Laging, Wesley J. Visser, Pedro Miranda Afonso, Jeroen GHP Verhoeven, Ingrid RAM Mertens zur Borg, Dennis A Hesselink, Michiel GH Betjes, Madelon van Agteren, David Severs, Jacqueline van de Wetering, Robert Zietse, Michel J Vos, Ido P Kema, Marcia ML Kho, Joke I Roodnat.

**Methodology:** Gideon Post Hospers, Mirjam Laging, Wesley J. Visser, Pedro Miranda Afonso, Jeroen GHP Verhoeven, Ingrid RAM Mertens zur Borg, Dennis A Hesselink, Anneke ME de Mik-van Egmond, Michiel GH Betjes, Madelon van Agteren, David Severs, Jacqueline van de Wetering, Michel J Vos, Ido P Kema, Marcia ML Kho, Joke I Roodnat.

**Project administration:** Gideon Post Hospers, Mirjam Laging, Wesley J. Visser, Jeroen GHP Verhoeven, Dennis A Hesselink, Michel J Vos, Marlies EJ Reinders, Joke I Roodnat.

**Resources:** Dennis A Hesselink, Anneke ME de Mik-van Egmond, Madelon van Agteren, Jacqueline van de Wetering, Michel J Vos, Marcia ML Kho, Marlies EJ Reinders, Joke I Roodnat.

**Software:** Gideon Post Hospers, Mirjam Laging, Dennis A Hesselink, Robert Zietse, Marcia ML Kho, Joke I Roodnat.

**Supervision:** Dennis A Hesselink, Marlies EJ Reinders, Joke I Roodnat.

**Validation:** Dennis A Hesselink, Anneke ME de Mik-van Egmond, David Severs, Joke I Roodnat.

**Visualization:** Ingrid RAM Mertens zur Borg, David Severs, Joke I Roodnat.

**Writing – original draft:** Gideon Post Hospers, Wesley J. Visser, Jeroen GHP Verhoeven, Ingrid RAM Mertens zur Borg, Anneke ME de Mik-van Egmond, Michiel GH Betjes, Madelon van Agteren, David Severs, Robert Zietse, Marcia ML Kho, Marlies EJ Reinders, Joke I Roodnat.

**Writing – review & editing:** Gideon Post Hospers, Mirjam Laging, Wesley J. Visser, Jeroen GHP Verhoeven, Ingrid RAM Mertens zur Borg, Dennis A Hesselink, Anneke ME de Mik-van Egmond, Michiel GH Betjes, Madelon van Agteren, David Severs, Jacqueline van de Wetering, Robert Zietse, Michel J Vos, Ido P Kema, Marcia ML Kho, Marlies EJ Reinders, Joke I Roodnat.

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
