## [Decision Letter · Decision Letter 0]

16 Dec 2024

PONE-D-24-50341

Pre-transplant residual diuresis and oxalic acid concentration influence kidney graft survival

PLOS ONE

Dear Dr. Roodnat,

Thank you for submitting your manuscript to PLOS ONE. After careful consideration, we have decided that your manuscript does not meet our criteria for publication and must therefore be rejected.

I am sorry that we cannot be more positive on this occasion, but hope that you appreciate the reasons for this decision.

Kind regards,

Mohamed E Elrggal

Academic Editor

PLOS ONE

*Comments from PLOS Editorial Office: *

*Please disregard the comments made by reviewer 1, who submitted comments for a different manuscript. The Academic Editor's decision is based on the concerns raised by reviewers 2 and 3.*

Reviewers' comments:

Reviewer's Responses to Questions

**Comments to the Author**

1. Is the manuscript technically sound, and do the data support the conclusions?

Reviewer #1: Yes

Reviewer #2: Partly

Reviewer #3: Yes

2. Has the statistical analysis been performed appropriately and rigorously? 

Reviewer #1: I Don't Know

Reviewer #2: I Don't Know

Reviewer #3: Yes

3. Have the authors made all data underlying the findings in their manuscript fully available?

Reviewer #1: No

Reviewer #2: No

Reviewer #3: No

4. Is the manuscript presented in an intelligible fashion and written in standard English?

Reviewer #1: Yes

Reviewer #2: Yes

Reviewer #3: Yes

5. Review Comments to the Author

Reviewer #1: The authors did a retrospective observational study of potential live kidney donors at their centre between 2007 and 2021 to analyse the causes of their decline

Abstract

Line 26, please mention the precent of kidney related conditions that led to donor decline

Line 27, please verify what is meant by medically complex donors and what led to change in your practice in view of your centre protocol. Please provide data supporting that as the results showed only a negative slope of eGFR of accepted donors over time

Line 29, change the sentence “decline in estimated glomerular filtration rate (eGFR) of accepted donors” to “accepting donors with lower eGFR”

Methods

Line 61, can you please mention details about your centre including name, city , average annual transplantation rate

Results

Line 151, change “was female” to “ were females”

Line 153, the authors mentioned no difference when comparing female and male first-degree related donors (p=0.66).

However, the table didn’t provide subdivision of first degree related donors into males and females

It is not clear what is the difference between table 1 and 2 regarding donor sex and relation to recipients. The authors compared declined and accepted donors regarding age, sex and relation to recipient so the denominator should be always the number of donors

Line 160, “(p.adj <0.001)” please correct the typo to (p <0.001).

Line 167, please provide more about your centre protocol , what is the age limit for donation

Figure 1A and 1B provide early the same date, you can keep only 1B as precents are provided

Line 214-216 please provide the statistical test used and the p value for comparison between males and females

Line 216-219, please do a statical comparison between different age groups and provide p value

Line 222, figure 3 A its about sex not age. Please correct this typo

Line 224-227 provide the p value for change in slope of mean age, eGFR, and BMI over time

Line 231, the authors said “The regression line represents the change in donor age per day” I think they meant per year to day

Line 232, , the authors said “To interpret the change per year, multiply the slope coefficient by 365” please clarify this point and provide the change in a table to make it easier for the reader

Reviewer #2: Methods:

• The setting of the study is not clearly defined

• Blood samples were taken prior to transplantations, but it does not state how long after last RRT session? It is mentioned in the introduction that oxalic acid levels decrease after dialysis and rebound after 48 hours. This will affect the measured level significantly and creates bias by time between last dialysis session and transplantation. This should be mentioned in the study limitations.

• “In patients scheduled for a graft biopsy within 3 months after transplantation, blood was taken on the day of the biopsy for determination of serum concentrations of oxalic acid and its precursors.” Why was this follow up level compared to eGFR at day 7 in figure 3, rather than to eGFR at time of biopsy. If this was not a protocol biopsy, then these patients were doing a biopsy for an assumed graft dysfunction, that may have affected the results. Then in the results section it is said that: “Almost all concentrations were obtained within 2 weeks after transplantation (median 7 (IQR 5-10) days after transplantation).” If this is the case, then it should be mentioned in the methods.

• There is no reference provided for the organic acid measurement methodology and for the used statistics.

Results:

• Results, tables and figures are disorganized.

• Table 1: unit of some items are not mentioned, e.g. BMI, median IQR in cold ischemia time.

• The categorization of the study cohort by residual urine volume and by oxalic acid levels are not clear and should be presented in a table or graph.

• Were there 3 categories of residual urine volume (<100, 100-1000 and >1000ml/day) or is it < 200 and > 2000 ml/day as in figures 2A and B, which denote the 25th and 75th percentile? The use of different threshold is a bit confusing.

• How many deaths occurred and were censored? It is mentioned in the patient survival section but should also be mentioned in the “Graft survival censored for death” section.

• “The cubic spline Cox regression model showed a linear relationship between residual diuresis and the log-hazard ratio of graft failure censored for death (graph not shown).” Better present it even if in a supplementary file as it is an important outcome

• Figure 4A and B are mentioned before figure 3 and are not present in the manuscript, or are they actually 2A and B?

• Consider omitting table 2.

• As figure 2 compares between living and deceased donor, why not add a table showing the results in each group regarding the oxalic acid levels, residual volume, death censored graft survival and patient survival?

• “In patients with a relatively good functioning kidney graft, oxalic acid concentrations had decreased to concentration below the upper limit of normal. Oxalic acid concentrations were still high in patients with a poorly or nonfunctioning graft (Figure 3).” What is meant by “relatively good functioning kidney” and “poorly or nonfunctioning graft”? This statement is too vague.

• Any cases of oxalate deposition in the biopsy? Were the patients with high post-transplant oxalate levels further investigated?

• “Variables with significant influence in univariable analysis were: recipient age, diabetes mellitus, concentrations of glyceric acid, pre transplant cardiac event, C-reactive protein (CRP) at day of transplantation, type of kidney replacement therapy, time on dialysis (months), body mass index (BMI), residual diuresis per day, previous vascular event, donor type, and donor age (data not shown).” Consider adding a supplementary table.

• “At 1 year after transplantation eGFR (estimated Glomerular Filtration Rate) was not significantly different between the patients with a functioning transplant with pre-transplant oxalic acid concentration ≤60 or >60 µmol/L.” Where are the levels and the significance of their comparison?

Discussion:

• The discussion is presumptive at points and over-interpret the study findings and therefore needs to be re-written.

• “It is conceivable that excretion of large amounts of these products by the newly transplanted kidney may cause toxicity and damage impairing kidney function.” Yes oxalate are tubulotoxic and may cause oxalate nephropathy, but there is no evidence that short term exposure post-transplant in the recovering graft may have a damaging effect?

• “This means that the pre-transplant patients with low or absent residual diuresis, are less vital than those with significant residual diuresis volume.” What is meant by “less vital”??

• “Sudden excretion of these products by the newly transplanted kidney may cause damage to the kidney transplant.” There is no evidence backing such an assumption, neither in literature nor in the study findings.

• Although residual urine is a suboptimal marker of residual renal function, the role of residual renal function on patient and graft survival should be further discussed.

• There should be a clear separation of limitations from the conclusion

• “Dietary and pharmacological interventions are defined to ensure optimal native kidney function preserving care[47].” This statement needs to be further discussed.

• Recommendations on how to preserve residual urine (and renal function) and lower oxalate levels in ESKD patients need to be separately discussed in a paragraph preceding the limitations. But are recommendations for patients with enteric hyperoxaluria appropriate for other ESKD patients and will they be as effective?

• A concluding paragraph should be added

Reviewer #3: The manuscript is well written and discussing Important issues and the following are few point to be considered by authors to improve the manuscript:

1- Please define Graft failure in the methods section (On which bases you decided that one recipient has graft failure?).

2- Please clarify when urine was collected for determinations of residual urine especially for HD patients as urine volume differs between dialysis and non dialysis days.

3- in Table 1 you mentioned that 20 had hyperoxaluria non renal causes, please clarify the cause of hyperoxaluria in those patients under or above the table. Has any of them had primary hyperoxaluria?

4- I think the relation between the Oxalic acid and eGFR day 7 after transplantation is better explained as follow: better eGfr lead to better excretion of oxalic acid so lower oxalic acid level not the reverse.

5- In your Conclusion you used your results to support pre-impetive transplantation, to do that you need to show us the difference between candidates who were not on dialysis and those who were on dialysis before transplantation regarding levels of oxalic acid and its metabolites , residual urine volume and outcomes (graft survival and patient survival), please add this comparison in result section.

6- figures need better adjustment (they are hazy and not very clear)

6. PLOS authors have the option to publish the peer review history of their article (what does this mean? ). If published, this will include your full peer review and any attached files.

**Do you want your identity to be public for this peer review?** For information about this choice, including consent withdrawal, please see our Privacy Policy .

Reviewer #1: No

Reviewer #2: **Yes: ** Yasmine Naga

Reviewer #3: **Yes: ** Sara T Ibrahim

- - - - -

---

## [Author Response · Author response to Decision Letter 1]

24 Jan 2025

Comments from PLOS Editorial Office:

Please disregard the comments made by reviewer 1, who submitted comments for a different manuscript. The Academic Editor's decision is based on the concerns raised by reviewers 2 and 3.

Reviewers' comments:

Reviewer's Responses to Questions

Comments to the Author

1. Is the manuscript technically sound, and do the data support the conclusions?

Reviewer #1: Yes Not applicable

Reviewer #2: Partly This is a technically sound prospective cohort study. “Experiments, controls, replications” are not applicable to this type of analysis. Population size is very large for this type of research. The conclusions are drawn based on the data presented and literature available.

Reviewer #3: Yes

2. Has the statistical analysis been performed appropriately and rigorously?

Reviewer #1: I Don't Know Not applicable

Reviewer #2: I Don't Know The statistical analysis has been performed under strict supervision of our statistician Pedro Miranda Afonso from the department of Epidemiology and Biostatistics. Because of his involvement in this study he is awarded as fourth author.

Reviewer #3: Yes

3. Have the authors made all data underlying the findings in their manuscript fully available?

Reviewer #1: No Not applicable

Reviewer #2: No Data are available on request.

Reviewer #3: No Data are available on request.

4. Is the manuscript presented in an intelligible fashion and written in standard English?

Reviewer #1: Yes Not applicable

Reviewer #2: Yes

Reviewer #3: Yes

5. Review Comments to the Author

Reviewer #1: The authors did a retrospective observational study of potential live kidney donors at their centre between 2007 and 2021 to analyse the causes of their decline Not applicable

Abstract

Line 26, please mention the precent of kidney related conditions that led to donor decline

Line 27, please verify what is meant by medically complex donors and what led to change in your practice in view of your centre protocol. Please provide data supporting that as the results showed only a negative slope of eGFR of accepted donors over time

Line 29, change the sentence “decline in estimated glomerular filtration rate (eGFR) of accepted donors” to “accepting donors with lower eGFR”

Methods

Line 61, can you please mention details about your centre including name, city , average annual transplantation rate

Results

Line 151, change “was female” to “ were females”

Line 153, the authors mentioned no difference when comparing female and male first-degree related donors (p=0.66).

However, the table didn’t provide subdivision of first degree related donors into males and females

It is not clear what is the difference between table 1 and 2 regarding donor sex and relation to recipients. The authors compared declined and accepted donors regarding age, sex and relation to recipient so the denominator should be always the number of donors

Line 160, “(p.adj <0.001)” please correct the typo to (p <0.001).

Line 167, please provide more about your centre protocol , what is the age limit for donation

Figure 1A and 1B provide early the same date, you can keep only 1B as precents are provided

Line 214-216 please provide the statistical test used and the p value for comparison between males and females

Line 216-219, please do a statical comparison between different age groups and provide p value

Line 222, figure 3 A its about sex not age. Please correct this typo

Line 224-227 provide the p value for change in slope of mean age, eGFR, and BMI over time

Line 231, the authors said “The regression line represents the change in donor age per day” I think they meant per year to day

Line 232, , the authors said “To interpret the change per year, multiply the slope coefficient by 365” please clarify this point and provide the change in a table to make it easier for the reader

Reviewer #2: Methods:

• The setting of the study is not clearly defined Settings are in lines 103-125 (In the manuscript with track-changes). Extra text is added.

• Blood samples were taken prior to transplantations, but it does not state how long after last RRT session? It is mentioned in the introduction that oxalic acid levels decrease after dialysis and rebound after 48 hours. This will affect the measured level significantly and creates bias by time between last dialysis session and transplantation. This should be mentioned in the study limitations. Time between last dialysis session and transplantation is a variable we studied (Table 1). The influence was significant in univariable analysis (supplementary table 2), but not significant in multivariable analysis. Of course concentration of oxalic acid is closely related to time between dialysis and transplantation. We were interested in the influence of the concentration of oxalic acid the newly transplanted kidney is exposed to. Though time did not have a significant influence, concentration of oxalic acid did.

• “In patients scheduled for a graft biopsy within 3 months after transplantation, blood was taken on the day of the biopsy for determination of serum concentrations of oxalic acid and its precursors.” Why was this follow up level compared to eGFR at day 7 in figure 3, rather than to eGFR at time of biopsy. If this was not a protocol biopsy, then these patients were doing a biopsy for an assumed graft dysfunction, that may have affected the results. Then in the results section it is said that: “Almost all concentrations were obtained within 2 weeks after transplantation (median 7 (IQR 5-10) days after transplantation).” If this is the case, then it should be mentioned in the methods. You are right, extra text is added to the Methods in lines 124-125.

• There is no reference provided for the organic acid measurement methodology and for the used statistics. Extra text on organic acid determination is added in lines 136-145.

Statistics were performed in SPSS in close collaboration with our statistician as mentioned in the text. The reference Gauthier (38) was added to the text in line 158 regarding the analysis using cubic splines.

Results:

• Results, tables and figures are disorganized. See below. Indeed, in line 235-236 figure 4A and 4B is mentioned that should be figure 2A and 2B. However, as currently 2 extra Figures were added, numbers of figures were adapted again.

• Table 1: unit of some items are not mentioned, e.g. BMI, median IQR in cold ischemia time. We added the missing items in Table 1.

• The categorization of the study cohort by residual urine volume and by oxalic acid levels are not clear and should be presented in a table or graph. It is known from the literature that there is a super saturation threshold concentration of oxalic acid: above this concentration oxalic acid crystallizes/deposits. This threshold is presumed to be within the range of 30-40 µmol/L (line 67-70). Extra text is added. The cubic spline Cox regression analysis indeed showed a non-linear relationship between oxalic acid concentration and graft survival censored for death (lines 208-209 and Figure 3). This is the reason we had to categorize the variable oxalic acid in 2 groups: ≤60 and >60 µmol/L. New text is added in line 212-252 to explain our choice. Evidently, measurable damage only occurs at concentrations above 60 µmol/L; above the supposed super saturation threshold of 30-40 µmol/L (lines 343-344).

For residual diuresis see answer to the next remark.

• Were there 3 categories of residual urine volume (<100, 100-1000 and >1000ml/day) or is it < 200 and > 2000 ml/day as in figures 2A and B, which denote the 25th and 75th percentile? The use of different threshold is a bit confusing. We describe <1000 and >1000 as clinical information, as many dialysis nephrologists are interested in these thresholds.

It is correct that in figures 2A and B (now 4A and B) we show the effect of residual diuresis volume of 200 and 2000 ml/day (25th and 75th percentile) on the graft failure risk.

We agree that it may be confusing. Information about the thresholds <1000 and >1000 is removed (lines 196-201).

• How many deaths occurred and were censored? It is mentioned in the patient survival section but should also be mentioned in the “Graft survival censored for death” section. Numbers of deaths were added to this section in lines 204-205.

• “The cubic spline Cox regression model showed a linear relationship between residual diuresis and the log-hazard ratio of graft failure censored for death (graph not shown).” Better present it even if in a supplementary file as it is an important outcome We added the cubic spline as supplementary figure 1A and B..

• Figure 4A and B are mentioned before figure 3 and are not present in the manuscript, or are they actually 2A and B? Yes, you are right. Apologies. Changes were made to line 235-236. However, as 2 Figures are added, numbers of Figures changes again.

• Consider omitting table 2. We removed table 2 and added it as supplementary table 2

• As figure 2 compares between living and deceased donor, why not add a table showing the results in each group regarding the oxalic acid levels, residual volume, death censored graft survival and patient survival? Figure 4 (was 2) not only compares the influence of living versus deceased donor type but also the influence of residual diuresis, and oxalic acid category in a population with the same donor age. Figure 4 is based on table 2: The final model of multivariable analysis. The only variables with significant influence were residual diuresis, oxalic acid category, donor type and donor age. In the multivariable analysis graft survival censored for death and patient survival is corrected for all variables presented in table 1. Extra text is added to the legend of figure 4 and in line 257.

A new versions of Table 1 was produced that shows information on living and deceased donor populations.

• “In patients with a relatively good functioning kidney graft, oxalic acid concentrations had decreased to concentration below the upper limit of normal. Oxalic acid concentrations were still high in patients with a poorly or nonfunctioning graft (Figure 3).” What is meant by “relatively good functioning kidney” and “poorly or nonfunctioning graft”? This statement is too vague. You can see that in figure 5: on the Y-axis kidney function (eGFR) and on the X-axis oxalic acid concentration. Text was rephrased in lines 290-294 to be more clear.

• Any cases of oxalate deposition in the biopsy? Oxalate depositions in the kidney are not part of routine examination in our center. In only 8 patients oxalate depositions in the biopsy were spontaneously reported. However, visible oxalate depositions are not obligatory for oxalic acid toxicity (see ref 26-29). Reference 29 is added.

Were the patients with high post-transplant oxalate levels further investigated? Patients with high post-transplant oxalic acid concentration were not separately analyzed, but part of the study. However, in line 297-299 we report that though eGFR was not different between the groups at 1 year after transplantation, the number of graft losses was far higher in the population with oxalic acid concentration >60 µmol/L. This is in line with results of graft survival censored for death. Our clinical experience is that all oxalic acid concentrations decrease as soon as there is a functioning kidney graft. Even moderate kidney function results in a decrease of concentrations.

• “Variables with significant influence in univariable analysis were: recipient age, diabetes mellitus, concentrations of glyceric acid, pre transplant cardiac event, C-reactive protein (CRP) at day of transplantation, type of kidney replacement therapy, time on dialysis (months), body mass index (BMI), residual diuresis per day, previous vascular event, donor type, and donor age (data not shown).” Consider adding a supplementary table. We removed the text in lines 267-270 and added a supplementary table 3 with these data.

• “At 1 year after transplantation eGFR (estimated Glomerular Filtration Rate) was not significantly different between the patients with a functioning transplant with pre-transplant oxalic acid concentration ≤60 or >60 µmol/L.” Where are the levels and the significance of their comparison? You are right, we added the information in lines 297-299.

Discussion:

• The discussion is presumptive at points and over-interpret the study findings and therefore needs to be re-written. The discussion is adapted based on the criticisms you describe below.

• “It is conceivable that excretion of large amounts of these products by the newly transplanted kidney may cause toxicity and damage impairing kidney function.” Yes oxalate are tubulotoxic and may cause oxalate nephropathy, but there is no evidence that short term exposure post-transplant in the recovering graft may have a damaging effect? In our article about DGF in Transplant International (ref 37) we show that the concentration of the direct precursor of oxalic acid, Glyoxylic acid, is associated with DGF. This is a new and revealing finding! Of course there is convincing literature that DGF results in worse long term graft survival. In the present manuscript, we are the first to show that oxalic acid concentration affects long term graft survival as well: Figures 4A and B show a negative effect of residual diuresis and oxalic acid directly post transplantation, but the lines diverge indicating a continuing negative effect on the long term.

There is no other transplant related evidence. However, Ethylene glycol poisoning often leads to (native or graft) kidney damage if untreated. Ethylene glycol is not kidney toxic itself, but its metabolites are kidney toxic: glycoaldehyde, glycolic acid, glyoxylic acid and oxalic acid. We showed that the kidney graft is placed in a toxic environment with high levels of glycolic acid, glyoxylic acid and oxalic acid. This comparison is added to the Introduction in lines 84-94. Figure 1 is added to show how ethylene glycol is metabolized to oxalic acid. References 32, 33 and 34 were added

• “This means that the pre-transplant patients with low or absent residual diuresis, are less vital than those with significant residual diuresis volume.” What is meant by “less vital”?? See the lines above this part: “ low residual kidney function is associated with serious comorbidities[27, 41-47].” We added extra text in lines 327-329. These differences are also evident in the dialysis population versus the pre-emptive population. We added a supplementary table 1 that shows characteristics in the populations on dialysis versus pre-emptiv

---

## [Decision Letter · Decision Letter 1]

14 Feb 2025

PONE-D-24-50341R1Pre-transplant residual diuresis and oxalic acid concentration influence kidney graft survivalPLOS ONE

Dear Dr. Roodnat,

Thank you for submitting your manuscript to PLOS ONE. After careful consideration, we feel that it has merit but does not fully meet PLOS ONE’s publication criteria as it currently stands. Therefore, we invite you to submit a revised version of the manuscript that addresses the points raised during the review process. Comments from PLOS Editorial Office: *We note that one or more reviewers has recommended that you cite specific previously published works. As always, we recommend that you please review and evaluate the requested works to determine whether they are relevant and should be cited. It is not a requirement to cite these works. We appreciate your attention to this request.*

We look forward to receiving your revised manuscript.

Kind regards,

Mohamed E Elrggal

Academic Editor

PLOS ONE

Journal Requirements:

"Funding: Foundation “Stichting de Merel”"

4. In this instance it seems there may be acceptable restrictions in place that prevent the public sharing of your minimal data. However, in line with our goal of ensuring long-term data availability to all interested researchers, PLOS’ Data Policy states that authors cannot be the sole named individuals responsible for ensuring data access (http://journals.plos.org/plosone/s/data-availability#loc-acceptable-data-sharing-methods).

5. We notice that your supplementary figure/tables are uploaded with the file type 'Other'. Please amend the file type to 'Supporting Information'. Please ensure that each Supporting Information file has a legend listed in the manuscript after the references list.

Additional Editor Comments (if provided):

Thanks for addressing the reviewers' comments. The manuscript appears much better now, with still some minor comments to address.

Reviewers' comments:

Reviewer's Responses to Questions

**Comments to the Author**

1. If the authors have adequately addressed your comments raised in a previous round of review and you feel that this manuscript is now acceptable for publication, you may indicate that here to bypass the “Comments to the Author” section, enter your conflict of interest statement in the “Confidential to Editor” section, and submit your "Accept" recommendation.

Reviewer #2: All comments have been addressed

Reviewer #3: All comments have been addressed

Reviewer #4: All comments have been addressed

2. Is the manuscript technically sound, and do the data support the conclusions?

Reviewer #2: Yes

Reviewer #3: Yes

Reviewer #4: Yes

3. Has the statistical analysis been performed appropriately and rigorously? 

Reviewer #2: Yes

Reviewer #3: Yes

Reviewer #4: Yes

4. Have the authors made all data underlying the findings in their manuscript fully available?

Reviewer #2: Yes

Reviewer #3: Yes

Reviewer #4: Yes

5. Is the manuscript presented in an intelligible fashion and written in standard English?

Reviewer #2: Yes

Reviewer #3: Yes

Reviewer #4: Yes

6. Review Comments to the Author

Reviewer #2: Most of the previously mentioned comments were adequately addressed. But a few changes still need to be done.

Introduction:

The introduction is a bit disjointed, and the paragraph need sentence openers to connect them.

Methods:

The organic acid measurement method is not described in detail but us not backed by any reference. Is the methos novel or has it been previously used and described in literature?

Results:

“There were 19 patients with enteric hyperoxaluria, one patient with primary hyperoxaluria was included.” Was the diagnosis made before transplantation or after. Did the patient with primary hyperoxaluria receive a liver transplant?

Always put a comma before “”respectively”

“In our population, the estimated turning point is at about 60 µmol/L in the recipient population with a living donor and at 70 µmol/L, in the recipient population with a deceased donor kidney” better replace the term turning point. Was it the level at which increased graft survival decreased?

Reviewer #3: (No Response)

Reviewer #4: I read with interest the revised version of this manuscript regarding the role of residual kidney function in limiting the role of oxalate toxicity after transplant. I only have some minor comments about how to improve the high quality of the manuscript.

- Despite not being reported and standardized as in the present study, we have the same clinical impression about patients with and without residual diuresis; patients without residual diuresis often showed extrarenal deposits of oxalate (e.g., in bone and soft tissues) and frequently experienced DGF, so we performed in some cases post-transplant dialysis in patients with not satisfying diuresis output and combined high oxalate levels. Do the authors have the same experience or suggest some strategies for patients at high risk for tubular toxicity (e.g., belatacept used to reduce combined CNI toxicity as in another clinical setting, see in example 10.1371/journal.pone.0240335)?

- At the same time, incremental dialysis and high-volume sessions (e.g., five times/week) in patients on active waiting list with progressive reduction of urinary output were suggested in our center to reduce the oxalate deposition. Have the authors suggested or implemented this strategy based on their results? Please comment on it.

7. PLOS authors have the option to publish the peer review history of their article (what does this mean? ). If published, this will include your full peer review and any attached files.

**Do you want your identity to be public for this peer review?** For information about this choice, including consent withdrawal, please see our Privacy Policy .

Reviewer #2: **Yes: ** Yasmine Naga

Reviewer #3: **Yes: ** Sara T Ibrahim

Reviewer #4: No

---

## [Author Response · Author response to Decision Letter 2]

26 Feb 2025

14-2-2025

PLOS ONE Decision: Revision required [PONE-D-24-50341R1] - [EMID:2259ed41066de6db]

PLOS ONE

Answers to Reviewers' comments:

Reviewer's Responses to Questions

Comments to the Author

1. If the authors have adequately addressed your comments raised in a previous round of review and you feel that this manuscript is now acceptable for publication, you may indicate that here to bypass the “Comments to the Author” section, enter your conflict of interest statement in the “Confidential to Editor” section, and submit your "Accept" recommendation.

Reviewer #2: All comments have been addressed

Reviewer #3: All comments have been addressed

Reviewer #4: All comments have been addressed

2. Is the manuscript technically sound, and do the data support the conclusions?

Reviewer #2: Yes

Reviewer #3: Yes

Reviewer #4: Yes

3. Has the statistical analysis been performed appropriately and rigorously?

Reviewer #2: Yes

Reviewer #3: Yes

Reviewer #4: Yes

4. Have the authors made all data underlying the findings in their manuscript fully available?

Reviewer #2: Yes

Reviewer #3: Yes

Reviewer #4: Yes

5. Is the manuscript presented in an intelligible fashion and written in standard English?

Reviewer #2: Yes

Reviewer #3: Yes

Reviewer #4: Yes

6. Review Comments to the Author

Reviewer #2: Most of the previously mentioned comments were adequately addressed. But a few changes still need to be done.

Introduction:

The introduction is a bit disjointed, and the paragraph need sentence openers to connect them.

Thank you for your efforts to improve our manuscript. Indeed, the Introduction improved significantly after textual adjustments to improve the connection in lines 60-105.

Methods:

The organic acid measurement method is not described in detail but us not backed by any reference. Is the methos novel or has it been previously used and described in literature?

Groningen is the reference lab for clinical chemistry and metabolic diseases for the northern half of the Netherlands since many decades. The method used is a further developed and improved method, based on previous methods since the seventies of the former century. The current method is currently being written into a manuscript for publication.

Extra text is added to line 141. The methods, on which the current method is based on, are referred in line 144.

Extra text on the internal quality control was added to lines 151-157.

Results:

“There were 19 patients with enteric hyperoxaluria, one patient with primary hyperoxaluria was included.” Was the diagnosis made before transplantation or after. Did the patient with primary hyperoxaluria receive a liver transplant?

Indeed an important clinical question. Extra text is added to lines 196-199 on this primary hyperoxaluria patient.

Always put a comma before “”respectively”

The text was adapted in lines 204, and 294.

“In our population, the estimated turning point is at about 60 µmol/L in the recipient population with a living donor and at 70 µmol/L, in the recipient population with a deceased donor kidney” better replace the term turning point. Was it the level at which increased graft survival decreased

Indeed, above this concentration the influence on graft survival is significant. Turning point was replaced by threshold for the influence on graft survival (line 220-221).

Reviewer #3: (No Response)

Reviewer #4: I read with interest the revised version of this manuscript regarding the role of residual kidney function in limiting the role of oxalate toxicity after transplant. I only have some minor comments about how to improve the high quality of the manuscript.

- Despite not being reported and standardized as in the present study, we have the same clinical impression about patients with and without residual diuresis; patients without residual diuresis often showed extrarenal deposits of oxalate (e.g., in bone and soft tissues) and frequently experienced DGF, so we performed in some cases post-transplant dialysis in patients with not satisfying diuresis output and combined high oxalate levels. Do the authors have the same experience or suggest some strategies for patients at high risk for tubular toxicity (e.g., belatacept used to reduce combined CNI toxicity as in another clinical setting, see in example 10.1371/journal.pone.0240335)?

Thank you for your compliment. Your description of extrarenal oxalate deposits and DGF in patients with low residual diuresis is really interesting and would perfectly fit into the experience we describe in the current manuscript. Do you have a protocol for post-transplant dialysis sessions in this population?

We do have a guideline for hyperoxaluria patients (primary or enteric) with high oxalic acid plasma concentrations: we try to decrease pre-transplantation oxalic acid concentrations and offer low-threshold daily dialysis sessions from day 1 onwards when urine production is below 1L per day in the first 24 hours after transplantation. Pre-transplant oxalic acid concentrations are determined only on indication: when primary or enteric hyperoxaluria is supposed. This means that this guideline does not hold for the entire transplant population.

We describe this guideline in our article from 2017 (reference 55) and are currently working on an update including 41 hyperoxaluria patients.

In our center Belatacept is seldom used to enable a decreased dose of CNI. We do not have a protocol for that.

- At the same time, incremental dialysis and high-volume sessions (e.g., five times/week) in patients on active waiting list with progressive reduction of urinary output were suggested in our center to reduce the oxalate deposition. Have the authors suggested or implemented this strategy based on their results? Please comment on it.

We do have this guideline, as described above, for patients with non-renal hyperoxaluria (primary or enteric). When pre-transplant oxalic acid concentration is above 60 µmol/L we suggest intensified dialysis sessions in the period before transplantation. It is left to the patient and treating nephrologist to what extent. In patients with a living donor dialysis is intensified (daily sessions) in the week before the transplantation. In patients awaiting a deceased donor offer dialysis can be intensified by switching to night dialysis or as you state: as many hours but in four to five times per week.

7. PLOS authors have the option to publish the peer review history of their article (what does this mean?). If published, this will include your full peer review and any attached files.

Do you want your identity to be public for this peer review? For information about this choice, including consent withdrawal, please see our Privacy Policy.

Reviewer #2: Yes: Yasmine Naga

Reviewer #3: Yes: Sara T Ibrahim

Reviewer #4: No

We successfully used PACE to convert and upload our graphs.

---

## [Editor Report · Decision Letter 2]

25 Mar 2025

Pre-transplant residual diuresis and oxalic acid concentration influence kidney graft survival

PONE-D-24-50341R2

Dear Dr. Roodnat,

We’re pleased to inform you that your manuscript has been judged scientifically suitable for publication and will be formally accepted for publication once it meets all outstanding technical requirements.

Kind regards,

Mohamed E Elrggal

Academic Editor

PLOS ONE

Additional Editor Comments (optional):

Thanks for addressing all the reviewers' comments.
---

## [Editor Report · Acceptance letter]

PONE-D-24-50341R2

PLOS ONE

Dear Dr. Roodnat,

I'm pleased to inform you that your manuscript has been deemed suitable for publication in PLOS ONE. Congratulations! Your manuscript is now being handed over to our production team.

Kind regards,

on behalf of

Dr. Mohamed E Elrggal

Academic Editor

PLOS ONE